# RESEKRA: Remote Enrollment Using SEaled Keys for Remote Attestation

**DOI:** 10.3390/s22135060

**Published:** 2022-07-05

**Authors:** Ernesto Gómez-Marín, Luis Parrilla, Gianfranco Mauro, Antonio Escobar-Molero, Diego P. Morales, Encarnación Castillo

**Affiliations:** 1Infineon Technologies AG, 85579 Neubiberg, Germany; gianfranco.mauro@infineon.com (G.M.); antonio.escobar@infineon.com (A.E.-M.); 2Departamento Electrónica y Tecnología de Computadores, Universidad de Granada, 18071 Granada, Spain; luis@ugr.es (L.P.); diegopm@ugr.es (D.P.M.); encas@ugr.es (E.C.)

**Keywords:** remote attestation, edge computing, Internet of Things, embedded systems, Trusted Platform Module

## Abstract

This paper presents and implements a novel remote attestation method to ensure the integrity of a device applicable to decentralized infrastructures, such as those found in common edge computing scenarios. Edge computing can be considered as a framework where multiple unsupervised devices communicate with each other with lack of hierarchy, requesting and offering services without a central server to orchestrate them. Because of these characteristics, there are many security threats, and detecting attacks is essential. Many remote attestation systems have been developed to alleviate this problem, but none of them can satisfy the requirements of edge computing: accepting dynamic enrollment and removal of devices to the system, respecting the interrupted activity of devices, and last but not least, providing a decentralized architecture for not trusting in just one Verifier. This security flaw has a negative impact on the development and implementation of edge computing-based technologies because of the impossibility of secure implementation. In this work, we propose a remote attestation system that, through using a Trusted Platform Module (TPM), enables the dynamic enrollment and an efficient and decentralized attestation. We demonstrate and evaluate our work in two use cases, attaining acceptance of intermittent activity by IoT devices, deletion of the dependency of centralized verifiers, and the probation of continuous integrity between unknown devices just by one signature verification.

## 1. Introduction

The number of things connected to the internet (Internet of Things, IoT) is growing exponentially, from 6.1 billion in 2018 to 14.7 billion in 2023 [1]. This growth is continuous, and there is no evidence that it will stop.

The current IoT structure relies on cloud computing. An IoT device collects information from the environment (sensor), sends it to a distant centralized server, which processes it along with currently stored information and information from other sensors, and generates, whether required, an action response. The response is then sent to a new IoT device, namely an actuator, which performs the appropriate action often in the nearby of the sensor. This is a simple and secure system from the service provider’s point of view, but due to the remoteness of the server from the nodes, it generates a high volume of data traffic and a high latency not suitable for real-time IoT applications. Moreover, it is not scalable enough with the current growth of IoT devices because of the large network use and the burden on cloud servers [2,3].

In this context, it is convenient for data to be processed at the edge for shorter response times, more efficient processing, and less pressure on the network. The technologies that enable computation at the edge of the network, on the data flow between the cloud services and the IoT services are known as edge computing [2]. In this scenario, the devices at the edge of the network that provide services (processing and/or data storage) are called edge nodes, while user devices are those that require real-time interaction or high storage capabilities. In this paradigm, user devices will act as edge nodes whenever possible, offloading their computing task to adjacent nodes if too burdensome at a given time [3].

On the other hand, the inherent distributed structure of edge computation implies additional security challenges over cloud computing. Egde computing has to protect different layer of technologies (from cloud to IoT devices) as in cloud computing, but it also needs to provide a distributed global connectivity of heterogeneous devices between the different layers. Rodrigo Roman et al. [4] define it as a combination of “the worst-of-all-worlds”. The scenario becomes more complicated when we consider that user devices continuously enter and leave the local network, that means, close mobile users (mobile subscribers), e.g., mobile phones or cars, which is called Mobile Edge Computing (MEC) [5].

In addition, it should be considered that the possible impact of an attack can be very high. Given the wide variety of situations in which edge computing can be used, the consequences can range from affecting our private information, the daily lives of users, and, more indirectly, the industrial ecosystem or critical infrastructure. It is therefore easy to override the possible benefits of edge computing by the losses that can result from relying too much on such technology [4]. Most edge devices do not have a user interface, which causes attacks to go unnoticed by most users [6]. Moreover, one of the main currently open problems in MEC is the security, in particular, the robustness of MEC servers [7]. Hence, verifying the status of IoT devices, edge nodes or not, becomes a complex and critical task in edge computing. This leads to the need of implementing a remote attestation system for edge devices and users [8].

In remote attestation, a Verifier checks the correct status of a device (Attestor) remotely. This solution has been widely investigated in the literature, but when applied to IoT systems, the current RA protocols are hard to scale. To overcome this challenge, several works recently proposed Collective RA (CRA) protocols. However, these solutions brought up new open issues [8]: the need of scalables and decentralized key management allowing mobility, the acceptance of intermittent activity of IoT nodes, which is essential for edge computing, and the resistance against the attack Time Of Check To Time Of Use (TOCTTOU) [9], which is barely covered in the State of the Art (SoA). In this paper, we present a solution for these problems through our remote attestation method, RESEKRA.

In our proposal, we attest remotely the state of an untrusted device through the support of a standard Root of Trust (RoT) consisting of a Trusted Platform Module [10]. This is a work already achieved and applied to edge computing in [11,12,13], but we go further.

Additionally to the results obtained in [11,12,13], we provide the following features:Proving authenticity in any communication also proves its correctness.No pre-shared secret is needed between the Verifier and the Attestor, which makes the system able to easily include new devices which is essential for systems with a variable number of nodes such as edge computing.Allowing secure software updates on the Attestor.Enabling offline remote attestation. The Attestor can be verified by the end user themselves.

For acheiving these features, we assume that:The Attestor shall include the needed hardware root of trust (RoT).A trusted Attestor manufacturer is available.The attacker cannot modify the RoT.

Our system, called RESEKRA, has been implemented on a Rapberry Pi by connecting a TPM and a pressure sensor to achieve a secure sensor acting as an Attestor that connects remotely to an external computer acting as a Verifier. The system attains an online and dynamic enrollment phase that works even on untrusted state devices, which allows it to be used in decentralized key management, where multiples entities are responsible for the key generation, distribution and regeneration with the advantage of fault tolerance, scalability [14] and flexibility. This makes RESEKRA the first remote attestation system for large networks of IoT devices with decentralized key management [8]. Moreover, Attestor can prove correct software status directly to the user devices, which make it perfect for users that just entered in the edge network, that means, mobile subscribers in MEC. Additionally, our scheme allows asynchronous booting, where the programs of the IoT devices are initialized in a random order, which is required for Linux-based systems, and finally, RESEKRA does not interfere with updates; therefore, a trusted online software updated system could be easily included in our system. To the best of our knowledge, there is no other system able to provide these services.

The rest of the manuscript is structured as follows: Section 2 is devoted to the related work, while Section 3 presents the technologies needed to understand the system. Section 4 details the RESEKRA design, whose implementation is presented in Section 5 as a solution for real scenarios. In Section 6, we will analyze the effects of the common attacks to the remote attestation systems in RESEKRA, and in Section 7, the conclusions of the work are carried out.

## 2. Related Work

Remote attestation is a process that has been studied extensively for many years and has recently focused on IoT nodes, creating their own field, Collaborative Remote Attestation (CRA) [8]. Due to the heterogeneity of IoT devices, there are several research studies of software-based solutions to increase scalability such as [15,16,17], but all require setting up one-hop networks between the Verifier and Attestor. This limitations make their implementation challenging. In addition, software-based solutions make strong assumptions about the limits of attacker capabilities, thus reducing their reliability [8]. To solve the latter problems, the use of a Root of Trust (RoT) residing in the device has been proposed in the literature [18]. This RoT is usually a mix of hardware and software. In [8], the authors divide RoT used in CRA into two main categories: those using hardware with the minimal security capabilities and those using a TPM.

In the branch of RoT based on hardware with the minimal security capabilities—also known as hybrid attestation techniques [19,20]—the authors specify their non-standard security hardware designs. SMART [21], ERASMUS [22], TrustLite [23] and SEED [20] provide high-level details of their hardware solutions. SEED and ERASMUS are the only solutions also requiring a Real-Time Clock, which is used to make the attestation to start from the Attestor itself. LISA [24] requires the same hardware architecture as SMART, but additional hardware is added to authenticate the Verifier and avoid DoS attacks. All the solutions mentioned above require the hardware design and manufacturing of novel security hardware, making them hard to implement. Another solution of the SoA is Hatt [19] where very limited additional hardware is required: “Physical Unclonable Functions” (PUF) and a ROM for the attention code. Still, they do not explain how to protect the PUF from being accessed by a malicious code. Finally, the research SARA [25] defines the requirements of their root of trust without providing any particular details. From all the presented works, only ERASMUS considers TOCTTOU attacks in their respective adversarial model.

On the other hand, other solutions choose security hardware with the highest security standards such as Trusted Plataform Module (TPM). In “Remote Enrollment using SEaled Keys for Remote Attestation” (RESEKRA), we opted for the use of TPM. As Tan et al. argue in “TPM-enabled Remote Attestation Protocol” (TRAP) [13], the cost of current TPMs is relatively low compared to the price of sensors, both in price and area. And as final product that can be found in the market, they can be easly implemented in real applications. As in the work proposed by Miguel Calvo and Marta Beltran [11] and MTRA (TRAP-based system) [12], we propose the use of a TPM where the Platform Configuration Registers (PCR) values, further discussed in Section 3.1.8, are used as proof of the current state of the IoT device. MTRA also considers TOCTTOU in their adversary analysis.

RESEKRA differs from all of the above works, whether based on hardware with the minimal security capabilities or TPM-based, in that none of them have any of the following features: enabling dynamic and online enrollment, which is essential for edge computing and even more for Mobile Edge Computing; decentralized key management and decentralized Verifier structure, removing single points of failure and trusted third parties; and correct analysis of devices with intermittent activity, which is critical in IoT computing because IoT devices are not always online. Additionally, we also consider TOCTTOU attacks, which are only covered in Erasmus and MTRA. All of them are open issues not addressed in the current Collaborative Remote Attestation solutions of the SoA [8].

## 3. Background

This section will outline the tools and technologies necessary for understanding RESEKRA: TPM2.0, Integrity Measurement Architecture (IMA) [26] and Core Root of Trust of Measurement (CRTM) [27].

These three elements are part of the Attestor. The CRTM ensures the correctness of firmware configuration and IMA. The IMA guarantees the correctness of the software configuration at runtime. Finally, the TPM2.0 ensures the reliability of all the information when shared externally (results generated by IMA and CRTM, among others). In Nomenclature part, we will summarize all the notations used in the document.

### 3.1. TPM 2.0

The Trust Platform Module 2.0 [10] provides standardized specifications for security coprocessors. Hereafter, security coprocessors that follow these specifications will be called TPM, and the device that has the TPM will be called the Host-TPM. These specifications have many details and functionalities, and only the features needed for our system will be described below.

#### 3.1.1. Virtual Memory

The TPM has a limited protected “real” memory, but credentials and sensitive data can be stored in the non-protected memory of the Host-TPM (computer that is using the TPM). This material is protected through signatures and encryption to ensure integrity and confidentiality. With this system, the TPM can use the Host-TPM device’s memory as a virtual protected memory of the TPM.

#### 3.1.2. Key Attribute: “Restricted”

The keys generated by the TPM have attributes that cannot be modified and which limit the use of these keys. Many of them must be verified to guarantee the security of the system. One of these essential attributes is Restricted. Keys with this attribute sign only TPM-generated digests in the signing process and will never sign a document starting with the value “0xff544347”, which is also called TPM_GENERATED.

#### 3.1.3. TPM_GENERATED

Data that begin with the code TPM_GENERATED can be signed by a Restricted key if and only if it was generated by the TPM. If the authenticity of the Restricted key is guaranteed, so is the veracity of the information related to the TPM. When used correctly, this allows much information to be remotely and reliably derived from the TPM in addition to the PCR values, such as the characteristics of the private keys stored in the TPM. This is a functionality that is rarely used in the state of the art and is the first reason why we stand out. With it, we can remotely complete the entire enrollment process of the IoT device to be attested.

#### 3.1.4. Policies

TPM2.0 offers a large set of policies that are not commonly used in typical applications. To the best of our knowledge, only [13] uses these available policies when sealing secret keys with PCR values. The use of these policies is one of the contributions carried out in RESEKRA. Only if the policies under which the TPM objects were created are satisfied, the object can be unsealed and then used. The three policies used in this work are the following: tpm2_policypcr: Seals the object to the value of one or more PCRs of the TPM.tpm2_policycountertimer: Seals the object to the number of times the TPM has been restarted.tpm2_policyauthorize: Seals the object to the policies signed by a Trusted Third Party (TTP). This policy allows changing the policies of a remotely signed object. With this policy, in RESEKRA, the Verifier is able to remotely update the other two policies, thus enabling flexible updating of the correct PCR value, and consequently, asynchronous booting and remote updates performing.

The name of an object (Object_name) is computed from the public information of the object (Object_Pubdata) as the public key (Object_PuB), the attributes, and policies. As a consequence, the Object_name can be used to assert the integrity of the object’s policies.

#### 3.1.5. Attestation Key

Attestation key (AK) is used to verify the internal information of the TPM. As mentioned above, one of its most important attributes is “Restricted”. The key name (AK_name) is computed from the public information of the attestation key (AK_Pubdata), as the public key (AK_PuB), attributes, and policies.

#### 3.1.6. Endorsement Key

The Endorsement Key (EK) is an assymetric key stored in the real memory of the TPM, which comes with a certificate signed from the TPM Manufacturer. The EK together with the Endorsement Key certificate (EK certificate) are used for two purposes: to verify the existence of the TPM and its manufacturer and to verify that an object is stored in the TPM. If an object, such as a private key, is shown to be stored in the TPM, its characteristics and attributes can be trusted as long as the TPM is trusted, such as the “Restricted” attribute.

It is a very restricted key and can only be used to decrypt data with a specific structure. Therefore, the TPM can prove ownership of the EK just through decryption operations [28]. For this task, we use the operations “makecredential/activatecredential”.

#### 3.1.7. Makecredential/Activatecredential

An external entity performs the *makecredential* operation using as inputs the Public Endorsement Key (*EK_PuB*), the name of a TPM object, usually the name of an attestation key (*AK_name)*, and a secret value:(1)Credential=makecredential(EK_PuB,AK_name,secret)

This command creates the so-called *Credential*. The TPM receives it and computes *activecredential* and, if and only if the AK_name object belongs to the TPM, it will decrypt the Credential by retrieving the secret. By showing knowledge of the secret, the Host-TPM proves that it has a TPM with EK and that the object AK_name is real and accurate.

#### 3.1.8. Platform Configuration Registers

The Platform Configuration Registers (*PCR*) are the basis for TPM-based remote attestation, which are also called Trusted Attestation Protocol (TAP) [29]. These registers can be updated through the “Extension” operation only [30]: (2)PCRnewvalue=hash(PCRoldvalue||datatoextend)

“Because of the one-way nature of a secure digest, there is no way to undo a measurement” [30]. Therefore, once a measurement has been stored in the PCR, no one, even with the highest privileges can override the effect of this measurement in the PCR, making PCRs perfect for verifying the integrity of the measurement list generated by the Integrity Measurement Architecture (IMA).

### 3.2. Integrity Measurement Architecture (IMA)

The Integrity Measurement Architecture (IMA) is a tool available in Linux responsible for measuring the files before they are accessed, storing the measurements in a list and extending them to the PCR. The IMA keeps a runtime measurement list; therefore, if any new or edited file is accessed, the path, name and content are measured, as shown in Equations (Equation 3) and (Equation 4), and they are included in the measurement list with the corresponding extension to the PCR.
(3)filedatahash=hash(filedata)
(4)measurement=hash(filedatahash,filepath|name)

Any program or file that would modify the system architecture such as the IMA itself would be measured before being executed, and it would leave an indelible mark on the PCR. Providing the measurement list together with the PCR will ensure the integrity of the measurement list. Therefore, for trusting in this attestation, it is needed to trust in the first program that started the measuring process and thus was never measured: the Core Root of Trust of Measurement.

### 3.3. Core Root of Trust of Measurement

As pointed out in [27], the Core Root of Trust of Measurement (CRTM) is the “first piece of BIOS code that executes on the main processor during the boot process. On a system with a Trusted Platform Module the CRTM (Core Root of Trust of Measurement) is implicitly trusted to bootstrap the process of building a measurement chain for subsequent attestation of other firmware and software that is executed on the computer system”. The CRTM is essential to being able to trust in IMA. The only way to trust in a CTRM is by trusting in the device’s manufacturer.

## 4. RESEKRA Description

As will be detailed in this section, the main contributions of RESEKRA can be summarized as follows:The use of Sealed Keys that can be used by the proprietary (the Attestor) temporarily and only when the current device status is approved by the Verifier, thus proving correct Attestor status only by proving ownership of the Sealed Key.The same Sealed Key can be used with different device status (always approved before-hand by the Verifier), allowing software updates without continuous revocations and avoiding a complex PKI.The creation of the Sealed Key can be realized completely remotely in a untrusted Attestor, allowing great flexibility and a plug-and-play business model perfect for Edge computing.

Figure 1 shows the general scheme of the remote enrollment process in RESEKRA. In the next subsections, the roles and operations performed in each of the steps will be detailed.

### 4.1. Roles

The main elements included in RESEKRA are the following:Attestor: edge devices with the hardware RoT, i.e., TPM and CRTM.Hardware Provider: the entity in charge of manufacturing the Attestor and providing the firmware and RoT. It is considered a trusted entity.Software Provider: the entity in charge of designing the Attestor software and creating the RML. It is a trusted entity.Programmer: the entity in charge of programming the Attestor. It is a not trusted entity.Deployer: the entity responsible for physically deploying the device and providing an authentication method. This entity is semitrusted, because it does not need a high level of knowledge to securely deploy the device, since there is no offline enrollment process and it has no interest in hacking the system because it is one of the stakeholders interested in providing the service. On the other hand, it would require a high level of expertise to manipulate physically the root of trust and go unnoticed by the rest of stakeholders.Verifier: the entity in charge of verifying the root of trusts of the Attestor, creating the SeK, verifying the Attestor status and authorizing the use of the SeK in the TPM.TPM: Hardware security module following the TPM 2.0 specifications.Edge computing user: the entity that requires communication with the Attestor to receive data (sensor) or to analyze/store data (Edge node).

In this paper, we do not focus on authentication. We consider that the Deployer is responsible for providing an authentication method for the Attestor, edge computing user and Verifier.

### 4.2. Certification of Manufacturer

In this first phase, the Attestor manufacturer and programmer shall create the needed certificates.

#### 4.2.1. Hardware Provider

When the Attestor is fabricated, the Device Manufacturer stores the necessary certificates (Manufacturer Certificates) in the device to identify the existence of the necessary root of trust (TPM and CRTM) on the Attestor. This certificate is signed by the Hardware Provider and linked to the EK. The Verifier can trust several Device Manufacturers.

#### 4.2.2. Software Provider

The software provider installs all the needed software in the Attestor and generates a list of measurements that will be used as reference, the Reference Measurements List (RML). This list is then signed by the software provider. The RML has to be provided to the Verifier and updated in case of software update. We will provide two possible solutions in Section 5. The software provider can be any stakeholder, e.g., Hardware provider, deployer or end user.

### 4.3. Remote Enrollment

At this moment in the process, the device can be deployed by the owner in any considered location. Because the rest of RESEKRA will work remotely, the deployment can be realized by inexperienced staff. Once the device is deployed, the Verifier has to assert the device’s hardware Root of Trust and to manage the creation and the validation of some special keys in the Attestor. It is a completely remote process and can be realized even in a untrusted Attestor. Therefore, the Verifier’s role can be dynamically changed, and even multiple Verifiers can work at the same time, sharing trust and responsibility.

#### 4.3.1. Edge Device Validation

Once the IoT device has been deployed, the Host_TPM requests the TPM for the public Endorsement Key (EK_PuK) and orders the creation of the Attestation Key (AK). It sends the public information of Attestation Key (AK_Pubdata) and the public Endorsement Key (EK_PuK) to the Verifier. The Verifier checks that the Endorsement Key belongs to a genuine TPM from a trusted manufactured edge device. The check is done through the Attestor’s Manufacturer Certificates. Then, it verifies the attributes of AK (the Restricted attribute, among others).

#### 4.3.2. Sealed Key Creation

The Verifier knows that the EK_PuK belongs to a trusted manufactured edge device but has not yet asserted that the Attestor is the owner of this EK. Then, the Verifier creates the asymmetric key pair, Authorizer, and encrypts the public key (Aut_PuK) with symmetric encryption using “Secret” as the symmetric key.

Next, it encrypts “Secret” using Makecredential and using EK_PuK and AK_Pubdata as input to create Credential. If the Host-TPM shows knowledge of Aut_PuK, it proves to own the TPM, EK, and AK with the asserted attributes. Finally, the Verifier sends Credential and Aut_PuK encrypted to Host-TPM. If Host-TPM shows knowledge of Aut_PuK, it proves the ownership of EK.

When Credential reaches Host-TPM, it decrypts the Credential with Activatecredential obtaining “Secret” and uses it to decrypt Aut_PuK. Now, because it has Aut_PuK, it can instruct TPM to create a sealed key (SeK) with the “tpm2_policyauthorize” using Aut_PuK as the authorizing key. The output of running the sealed key creation script is shown in Figure 2.

#### 4.3.3. Sealed Key Validation

The TPM generates an attestation certificate for SeK (Cert_SeK). This certificate is asserting that the object with the name SeK_name is stored in the TPM; hence, the TPM is bearing it. Then, the TPM signs it with the AK and sends it to the Verifier together with SeK_Pubdata.

The Verifier has to validate this information once received. First, the Verifier checks that the certificate starts with TPM_GENERATED and is signed with the suspected AK. This means that the TPM has generated the certificate if AK is a real AK, which will be verified in the next step. Finally, the Verifier needs to assert that this is the correct type of certificate.

It starts with TPM_GENERATED;It is signed by AK;It is an attest certificate.

This validation means that Cert_SeK is genuine. Now, the Verifier will recompute the SeK_name locally from SeK_Pubdata. It shall match with the name backed by the Cert_SeKcheck, and finally, the Verifier will verify all the features of SeK:The sealed key is the same being attested by the certificate;The policy used to create the sealed key is correct (tpm2_policyauthorize using Aut_Pub). This point is essential, because it proves knowledge of the Aut_Pub;Verifying the attributes of the SeK.

If all verifications are successful, the Verifier has been able to create completely remotely, without the need for pre-shared secret, a sealed key in the Host-TPM’s TPM. The Verifier would have complete control over this sealed key. The Attestor will be able to use this key only when satisfying the signed requirements imposed by the Verifier.

### 4.4. Attestation

At this point, the CRTM measures the BIOS and the IMA measures the integrity of all files executed. This is a runtime measurement, so when any new or modified file is executed, it is measured, and the result is stored in the measurement list and extended to the PCR.

The Attestor initiates the remote attestation by sending a request to the Verifier, and the last one replies by sending back a random nonce to the Attestor. The TPM generates a Quote signed by the AK, which includes the nonce to avoid replay attacks. The Quote is a certificate including several important parameters; those most relevant for RESEKRA are:TPM_GENERATED, to confirm the veracity when signed by the AK;The random nonce, to avoid replay attacks;Reset value of TPM. Value that changes when Host-TPM is rebooted;Value of PCR.

The Attestor sends the Quote together with the measurement list to the Verifier. It asserts the veracity of the certificate (TPM_GENERATED, signature, and random nonce) and uses the measurement list to rebuild the PCR value found in Quote. If the reconstruction is correct, it means that the measurement list was not adulterated.

Finally, the Verifier compares the measurement list with the Reference Measurement List (RML). The Verifier may have obtained this RML through the Host-TPM signed by the trusted Device Manufacturer or from the cloud. Now, the Verifier can check program by program if there is a difference and where.

When the list is approved by the Verifier, the Verifier can create an authorization to use the SeK for the particular values of RESET and PCR specified in Quote, and it sends the authorization signed by AuTK to the Host-TPM. Figure 3 shows the particular code to generate and sign the combined policy. The complete codes are accessible at: [31].

Once the authorization is received, the Host-TPM sends it to the TPM to unlock the SeK. The TPM verifies the signature and checks that it meets the requirements of the authorization (RESET and PCR values), being able to use the SeK until these requirements change, i.e., until it reboots or until a new or modified program is used. If the PCR value changes, the Attestor will request a new authorization from the Verifier; however, if a program that was not supposed to be executed is run, the Attestor will not obtain a new authorization until it is restarted and ask for a new one. A device will not be able to reuse an old authorization because those are sealed to the RESET value of the TPM, which prevents the use of authorizations for old decommissioned software.

### 4.5. Daily Life Functionality

Whenever the egde device proves ownership of the SeK, the TPM will first verify the signature of the policy provided by the server, secondly, it will assert the signed polices (values of PCR and Reset), and finally, it will grant access to sign with the SeK, proving the correctness of the software, and therefore, no additional attestation is needed. Figure 4 shows the script using the SeK to sign the measure of a pressure sensor satisfying the policy that was set when the SeK was generated in Section 4.3.2 (authorization policy in Figure 2).

This key can also be used to prove authentication. In protocols such as the well-known cryptography protocol Transport Layer Security (TLS) [32], or signing transactions for blockchain, asymmetric keys are used for the authentication, thus; the Sealed key pair can be use to prove correctness at the same time that authentication. The end user can trust in the SeK because it is coming with a certificate from a trusted third party, which is commonly the Verifier itself.

### 4.6. Implementation

For implementing our system, we employ a low-price System on Chip (SoC), Raspberry Pi 4B: Broadcom BCM2711, Quad core Cortex-A72 (ARM v8) 64-bit SoC @ 1.5 GHz 8 GB LPDDR4-3200 SDRAM and a pressure sensor DPS310 Pressure Shield2Go [33]. Additionally, we have used a TPM IRIDIUM9670 TPM2.0 LINUX [34], a hardware security module by Infineon Technologies GMBH specifically designed for Raspberry Pi SoCs. In Figure 5 is shown the setup. Our implementation lacks CTRM, but it is a feasible solution for the real-world market [35].

The software has been developed over WenXin’s code, which is available in a public repository [36]. This software presents a classic remote attestation with Attestor and Verifier. We have used this repository as a baseline for the implementation of our presented novelties, and the result can be found in [37] for the Attestor and [31] for the Verifier.

## 5. RESEKRA Use Cases

There are several ways to use RESEKRA, depending on whether the edge device programmer is the same as the manufacturer, regarding how the Verifier obtains the RML, how it obtains the trust on the sealed key, or who takes the role of the Verifier. In the process, we consider that there is a trusted system to provide the identity of the ED devices to the end user. Two use cases were considered for RESEKRA: edge computing as a trusted service and edge computing as a trusted service—No TTP online.

In the case of edge computing as a trusted service, we have the case of a company that offers edge computing for a fee. This company is responsible for the deployment and maintenance of edge nodes. Additionally, in this use case, we can find the manufacturers of the edge devices, which are trusted by all. The Verifiers, in this case, are a cloud service with high latency and some downtime. Finally, we find the end users who rely on the Verifiers to use the reliable edge device services requiring very low latency.

The second use-case, edge computing as a trusted service—No TTP online, is similar but with a fundamental difference: there are no cloud services offering the Verifier service.

There are other possible use cases where the Reference Measurement List is trusted through a voting method or is provided along with the edge device software from a public repository such as [38]. Edge devices that have already been validated can also be used as a Verifier.

### 5.1. Edge Computing as Trusted Service

In edge computing as a trusted service, we have (1) the trusted entity in charge of designing, programming, and manufacturing the edge devices for edge computing (Trusted Designers); (2) several semi-trusted investors in charge of deploying and maintaining the edge devices (Semitrusted Maintainers); (3) Trusted Clouds and (4) mobile end users, e.g., a car.

#### 5.1.1. Roles

The Trusted Designers are responsible for designing an edge device with CRTM and TPM. They are also responsible for designing the secure software. Finally, they manufacture the product and store all necessary certificates on the edge device.

Semitrusted Maintainers are entities with little technical knowledge. They are semi-trusted because they have no interest in making attacks into the system, but they have no knowledge of how to prevent them. They have the capacity and interest to deploy and maintain an edge device in a specific area, such as a non-profit interested organization, e.g., a government, or an interested profit organization with a business model based on subscriptions or blockchain using utility tokens, e.g., Helium [39].

The Semitrusted Cloud (SC) is responsible for verifying and signing the authorizations for the SeKs of edge devices. Every end user has a set of trusted SC. The SC is semitrusted because the end user only trusts some of them.

The last role is the end user, who needs to trust edge devices to use edge computing services.

#### 5.1.2. Process

First, Semitrusted Maintainers purchase an Edge-Device N ($ED_{N}$) ∀N∈N from Trusted Designers and deploy it wherever they see a fit. Thanks to the remote and versatile key creation system in RESEKRA, there can be several independent SCs verifying the integrity and authenticity of the received documents. These SCs can belong to third parties. The $ED_{N}$ communicates with a set $O_{N}$ of SCs, sends them the reference measurement list and the manufacturing certificates signed by the Trusted Designers. A subset $P_{N}$ of $O_{N}$ approves the Section 4.3 Remote Enrollment and issues a certificate with the SeK generated and $ED_{N}$’s identifier. The identifier has to be an universally unique identifier [40] to avoid Wormhole attacks [41], which is explained in more detail in Section 6.3. In our scenario, we propose using the last 128 bits of the hash of the EK_PuB as an identifier.

Because the process Section 4.3.1 Edge device validation has to be realized just once per Verifier, and the processes Section 4.3.2 Sealed Key creation and Section 4.3.3 Sealed Key validation are performed just in unusual moments, normally after each reset, the interactions between $ED_{N}$ and SCs are infrequent. Therefore, the number of Verifiers in the subset $P_{N}$ will not affect the service quality. Moreover, the SeKs are stored in the virtual memory of the TPM.

The edge device subsequently requests a Section 4.4 Attestation to every SC in $P_{N}$. A subset $Q_{N}$ of $P_{N}$ approves the attestation and grants the authorization to use the SeKs, as long as the device is not rebooted or its software is not modified. Additionally, a time limit can be added to this authorization.

At this moment, the $ED_{N}$ can act as a Verifier for the nearest IoT devices by using a software-based remote attestation such as SoftWare-based ATTestation (SWATT) [15], where the time response is essential, and therefore, a one-hop network is a fundamental requirement. We do not provide further details here, since it is beyond the scope of this paper.

Finally, end users, by requesting the services of $ED_{N}$, have to provide a priority organized set *G* of SCs and a Random Nonce (RN). If any of the elements of *G* belong to $Q_{N}$, the $ED_{N}$ will sign the RN using the SeK approved by the most priority SC belonging to *G* and $Q_{N}$, and it will furnish the corresponding certificate. Since the identity in the certificate is based on the EK_PuK, the identity provider just has to provide one identifier for $ED_{N}$, and it is valid for all the $ED_{N}$’s SeKs. Otherwise, the $ED_{N}$ is considered untrusted by the end user.

When $ED_{N}$ is trusted, the end user requests the edge computing services (gathering data or processing data) and measures the required time of the service. At the end of the service, the $ED_{N}$ signs the hash of the RN with the results of the service using the SeK. If the final signature never arrives or takes longer than expected, the end user does not trust the results and reports the irregularity.

However, it needs to check the correctness of the edge devices before using their services. The car only has the name of the edge devices and a set of cloud services that are trusted for this car but are not accessible in real time.

Figure 6 shows an scenario where a car enters in a location with an edge computing network, and it needs edge services from 3 EDs such as gathering data from sensors or using computer processing. However, it needs to check the correctness of the edge devices before using their services. The car only has the identifier of the edge devices and a set of SCs *G* that are not accessible in real time.

Firstly, the car requests a signed RN from the three EDs, each one with three different subsets *Q*, but only $ED_{1}$ and $ED_{3}$ have passed through a remote attestation of at least one of the SCs belonging to *G*. Then, by simply signing the RN with their SeKs, $ED_{1}$ and $ED_{3}$ prove their correctness, and the car puts the trust in them and starts using their services.

### 5.2. Edge Computing as Service—No TTP Online

In edge computing as service—No TTP online, we consider the need of avoiding the use of a Semitrusted Cloud. Perhaps, because the internet connection is not available, there are not SCs of *G* belonging to *Q* or simply to avoid the use of a TTP.

In this field, we have the same roles as in the previous case, except for the absence of SC.

#### Processes

When the end user wants to use the services of the edge device, and there is not any trusted third party available to act as a Verifier, the end user himself can act as a Verifier.

The end user himself realizes the Section 4.3 Remote Enrollment. Then, the ED provides the RML signed by the ED’s programmer. If ED passes the Attestation Section 4.4, the end user will authorize the use of the SeK and requests the Edge computing services (gathering data or processing data). Upon completion of the service, the ED will use the SeK to sign the service result.

Throughout the communication process, a constant status of the software is ensured by continuously proving the SeK ownership. The process of document verification and SeK creation should be created beforehand to avoid delaying the start of the edge computing service. Figure 7 represents the differences between the two use-cases presented in this section. On the right of the image, all the phases (in purple) are processed between car and edge device, and there is no need of third party acting as a Verifier.

## 6. Security Analysis

In this section, we provide an analysis of the most common attacks on remote attestation present in the state of the art. We do not consider Denial of Service in all the analysis.

### 6.1. Man-In-the-Middle Attack (MIM Attack)

The entire RESEKRA scheme was developed with MIM attacks in mind. The most sensitive section of RESEKRA is Section 4.3 Remote Enrollment, since we consider the Host-TPM untrusted for the remote communication.

All communications between the ED and Verifier are protected with TLS, but we take into account that even in this communication, an MIM can obtain complete control of Host-TPM. All comunication between the Verifier and TPM goes through the Host-TPM. The attacker would be able to read all certificates and information transferred, but it would not be able to modify it, because it is signed by the TPM. Moreover, even with the complete control of the Host-TPM, the attacker cannot force the TPM to sign fake certificates or quotes because all the TPM generated data are signed by a restricted key (AK).

The AK is proved to be a restricted key by using the Endorsement Key. The attacker could obtain knowledge of Auth_PuK and generate a fake SeK out of the TPM and store it in the TPM. Following this process, the attacker could generate a Cert_SeK signed by the AK, but it would fail while asserting the SeK attributes Section 4.3.3. Therefore, all Section 4.3 Remote Enrollment is protected from MIM attack.

### 6.2. Impersonation and Replay Attack

An attacker could obtain the certificates stored in Host-TPM, but it would not be able to prove EK ownership and thus would fail to perform the Section 4.3 Remote Enrollment.

Validation of AK and SeK is done through Auth_PuK, which is randomly generated. Therefore, we avoid replay attacks in Section 4.3 Remote Enrollment.

Note that Section 4.4 Attestation is realized with an RN and signed by AK; therefore, replay attacks and impersonation are avoided.

Finally, to show the correct status of the software to end users, the SeK is used to sign RNs, avoiding replay attacks.

### 6.3. Wormhole Attack

This attack was introduced by Yih-Chun Hu [41]. In this attack, the attacker uses legitimate data or information from an edge computing network at a given location and “tunnels” it in some way to another edge computing network without adding appreciable delay. The attack will then be explained, which is followed by a possible defense strategy:1.The attacker compromises one edge device close to the end user ($ED_{C}$).2.The attacker tunnels the network of $ED_{C}$ with another far network with an ED from the same deployer ($ED_{f}$).3.The end user starts the authentication method with $ED_{C}$ and passes through because it is talking with the expected node.4.When the attacker receives a random nonce to be signed with SeK, it tunnels this random nonce to $ED_{f}$, obtains the signature and certificates $ED_{f}$’s SeK, and sends them back to the end user.5.Finally, the end user trusts $ED_{C}$ without access to $ED_{C}$’s SeK.

Since RESEKRA relies on asymmetric cryptography (EK, AK, and SeK) that is randomly generated and is therefore unique, it is easy to avoid this kind of attack.

To avoid this attack, the certificate issued from SCs shall include the universally unique identifier [40] (ID) of $ED_{C}$, and this ID should be provided to the end user as part of the $ED_{C}$’s identification information. In Section 5.1, we use the last 128 bits of the hash of the EK_PuB as the ID.

When verifying the SeK’s certificate, the end user will find out that the SeK of $ED_{f}$ does not belong to $ED_{C}$.

### 6.4. Interference Attacks

In the interference attack, an MiM provides a false attestation response (Quote) to Verifier. It will not pass the process, and ED will be considered untrustworthy, even though the Verifier was unable to verify the actual Quote.

The Quote is coming with a signature and RN. When a false Quote is provided, the Verifier will recognize it is not coming from the real ED and will refuse it without affecting the attestation process.

### 6.5. Time-of-Check-to-Time-of-Use Attack

Time-Of-Check-To-Time-Of-Use (TOCTTOU) is a present vulnerability in Remote Attestation scenarios [9]. In TOCTTOU, the ED provides evidence of having the appropriate software at the time of the Attestation, but it uses different software when the ED is about to provide services.

At a general level, this attack should be avoided by proving access to the SeK at the moment ED provides the services.

One possible attack would be to have the correct software when proving ownership of the SeK at the beginning of the service and then calling a manipulated program when providing the real services. This attack should not be possible in an IMA that is bugs-free because any program able to call a manipulated program would not pass the attestation.

However, if in an unlikely situation were to happen, the ED will not be able to provide the second signature of the result, and thus, the end user will notice this problem when receiving the result. If so, the ED owner can initiate a fast debugging process.

### 6.6. Verifier-Based DoS Attack

In Verifier-Based DoS (VBDoS) attack, the attacker acts as a Verifier and requests many attestations to the ED, which becomes overloaded and cannot provide its normal service.

The remote enrollment and attestation from ED to SC is started by the ED, and they are not routine procedures.

The attack could come from the end user when it is in charge of attesting the ED through the full RESEKRA process or just validating the use of the SeK. The solution for this attack should be provided by the ED owner as part of the normal DoS attacks to their EDs. It is the responsibility of the ED owner to develop a scheme for providing ED services to the end user, allocate resources, and control and avoid misuses or malicious uses of the ED resources by the end user.

### 6.7. Non-Verification DoS Attack

It is a new kind of attack developed by us. In this attack, the attacker is able to interrupt the Verifier service on an edge computing network. It could happen through attacking one of the hops in the connection between the Verifier and Attestor or with a DoS attack to the Verifier.

In a common remote attestation scheme, throughout the interruption, the end user cannot trust the EDs, and even when the EDs are in a correct state, if trust is essential, the attacker produces a DoS to the local edge computing infrastructure.

Due to the decentralized Verifier infrastructure in RESEKRA, a DoS to multiple Verifiers is very unlikely. If the connection between EDs and Verifier is broken, the ED still has access to the SeK; therefore, it can still prove its correct status. If by an unlikely circumstance, the ED loses access to the SeK, several local Verifiers can be run-time assigned, or the end user itself can perform the remote attestation. To the best of our knowledge, our system is the first remote attestation protocol able to resist this attack.

## 7. Conclusions

In this paper, a new process for remote attestation focused on edge computing has been presented. The process, called RESEKRA, has been designed and implemented in a real sensor, introducing new features with regard to the SoA. The first characteristic is the remote enrollment, when an IoT device can connect with a new Verifier without having to know a shared secret, allowing Attestors and Verifiers to dynamically join and leave the structure. The second special feature of RESEKRA is the use of asymmetric keys sealed to the correct state of the software, which allows the correct state of the device to be proven directly to the end user while reducing the workload on the Verifier and allowing the IoT sensor to operate with intermittent activity by requesting a remote attestation when it deems it necessary.

These qualities enable the sought-after decentralized verification system in the SoA, integrate remote attestation into Mobile Edge Computing, and finally, allow the end user to trust the edge devices without any trusted authority or entity within the edge. It allows for further research into edge computing by taking for granted the trust of the devices in the environment. Furthermore, RESEKRA can also have a significant academic impact on other lines of research in IoT computation where trust in IoT devices is crucial, e.g., supply chain, IoT in blockchain, or Industry 4.0.

On the other side, RESEKRA—as with all Remote Attestation systems based on RoT—requires a reliable manufacturer to produce the trusted device, but then, RESEKRA can be used in many other scenarios inside and outside Mobile Edge Computing. It can be used from applications as simple as smart homes—where a simple set up is essential—attesting the end users’s devices using a mobile phone. Other applications as complex as Industry 4.0 use several Verifiers from the same owner to avoid the highly feared DoS attacks in manufacturing affairs. It can also be used for lower consumption applications, where sensors operate with intermittent activity to save battery life. In general, RESEKRA and therefore its results can be applied in any scenario beyond the use cases presented where the assumptions detailed in the introduction are met.

In other respects and future work, we envision integrating the Direct Anonymous Attestation Scheme. It would allow proving ownership of an SeK without giving any additional device identity information. This would preserve the privacy of the Attestor and enable the attestation of privacy-sensitive devices such as cars or mobile phones. On the other side, we are currently working to delete the use of private keys in the Verifier and move it to a smart contract in the blockchain, making the blockchain itself being able to attest the IoT devices before accepting a transaction.

## Figures and Tables

**Figure 1 sensors-22-05060-f001:**
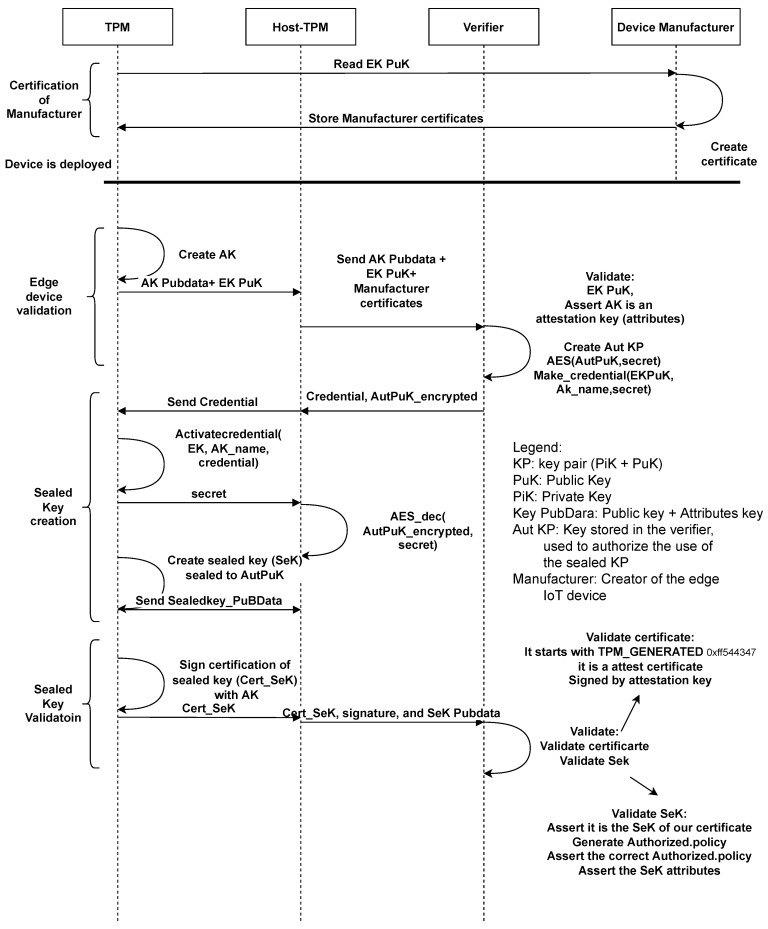
Remote enrollment process. The scheme represents all the enrollment process from the device manufacturer to the supervised creation of the sealed key.

**Figure 2 sensors-22-05060-f002:**
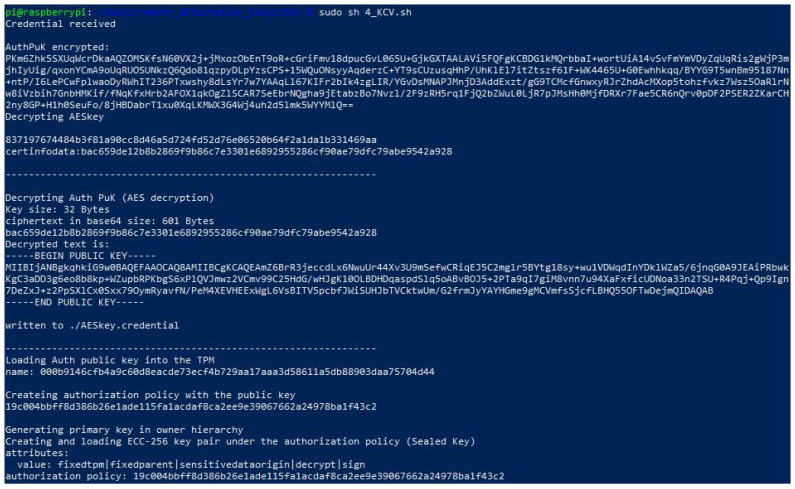
Output of the script for sealed key creation.

**Figure 3 sensors-22-05060-f003:**
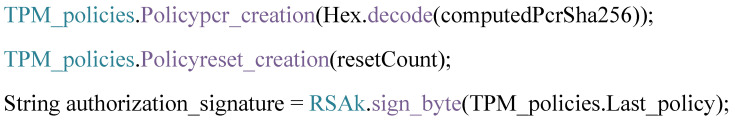
Section of the Verifier code where it computes the policies for using the SeK and signs it if and only if the Attestor passed through the attestation process.

**Figure 4 sensors-22-05060-f004:**
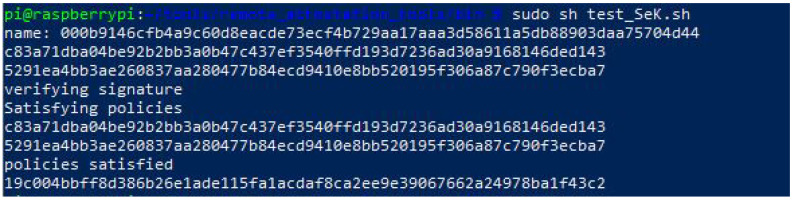
Output of the script using the sealed key to sign a measured value after satisfying the policies.

**Figure 5 sensors-22-05060-f005:**
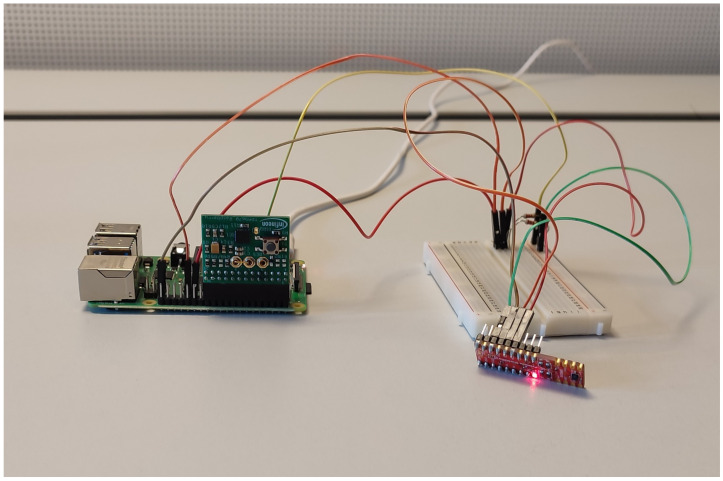
Experimental setup. Raspberry Pi 4 with TPM (green hardware) and pressure sensor (red hardware).

**Figure 6 sensors-22-05060-f006:**
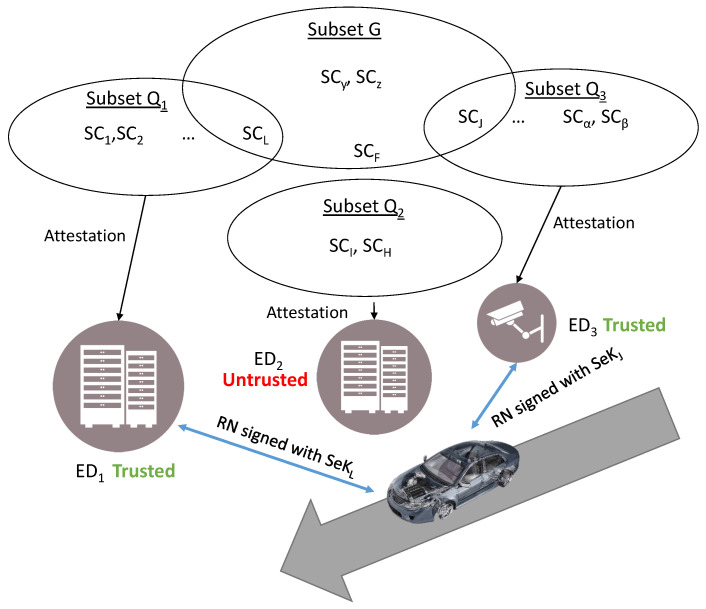
Visualization of edge computing as trusted service. A car reaching a location needs the local services of several edge and IoT devices. All these devices have many SeKs available validated by an SC each. The car just trusts in some of the SCs, those which belong to subset G.

**Figure 7 sensors-22-05060-f007:**
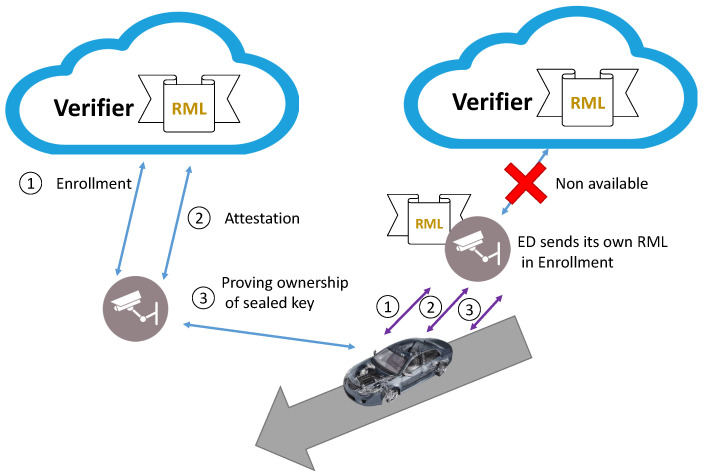
Visualization of the two use cases, edge computing as service (in blue) and edge computing as service—No TTP online (in purple).

## Data Availability

Not applicable.

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
