# Peer review of "RESEKRA: Remote Enrollment Using SEaled Keys for Remote Attestation"

_sensors, 2022, doi:10.3390/s22135060_

Round 1

Reviewer 1 Report

Security of autonomous systems which rely on edge devices requires the security of such devices. In this manuscript, the authors proposed a novel solution called RESEKRA that introduces new features.

The authors described the motivation and importance of cloud computing and edge nodes in Mobile Edge Computing. However, a gap in the article is noticed that the transition between motivation and the actual problem is not straightforward. The authors could improve the paragraph in lines 58-62. The contributions of this article are promising.

The background section helps the reader understand the necessary concepts to follow in the next sections. Figure 1 is the combination of different phases and the detailed steps in each stage, and it is easy to follow. There is a need for explanation in lines #370-372, the key used to prove authentication or sign transactions for the blockchain.

The authors implemented the RESEKRA system using Raspberry Pi and defined use cases to evaluate the system. The justification for selecting the two use cases is provided. It would be better if the authors show the difference between edge computing as a trusted service and edge computing as a trusted service – No TTP in a Figure with the roles.

The authors claimed this is the first protocol incorporating remote enrollment and guaranteeing device status correctness through classic secure communications, while allowing software heterogeneity and secure updates. In such cases, they need to write how the results of evaluated use cases are generalized beyond this study.

Minor: A table or glossary of acronyms and abbreviations would be helpful for the reader. 

Reviewer 2 Report

This paper proposed Trusted Platform Modules (TPM) and Core Roots of Trust for Measurement (CRTM) for creating a remote attestation protocol achieved and applied to Edge computing. However, there are many problems in this work.

1. there is on method with sensors in Edge computing with this work.

2. the related work is not good.

3. equations in this work are error. Eq. (1)(3) are not equation.

4. I have never seen any ‘computing’ processing in this work.

5. how about simulation or experiment in this work, please give readers a conclusion.

Reviewer 3 Report

The authors present a very interesting and pertinent paper. The content of the paper is very well aligned with the scope of the journal. It is well organized and very well detailed. Congratulations! Some points still that could improve the presented work:

1-     The abstract is not capturing all the value of the paper. It should be reformulated to highlight the most important findings and novelties of the proposed new process. Being this a paper of the natural sciences (extremely very technically), by this order the abstract should include: 1- the problem(s), 2-the implication(s) of the problem(s), 3-the solution(s) proposed to solve or tackle the problem(s), 4-the main result(s) of the solution (if a case study is available). I suggest the authors to get inspiration on the following paper to structure the abstract to capture the reader at the first blink: A Design Science Research Methodology for Information Systems Research (DSRM)

Ken Peffers, Tuure Tuunanen, Marcus A. Rothenberger Samir Chatterjee

https://www.tandfonline.com/doi/abs/10.2753/MIS0742-1222240302

2-     Subchapter 5.2.1 makes no sense unless there is a subchapter 5.2.2, which is not to be found in the document. I suggest that the content of 5.2.1 should be changed under 5.2.

3-     For such a well detailed and organized paper as from chapter 1 to chapter 6, the conclusions chapter is very poor. It should be discussed the implications (academic and managerial) of the proposed new process (RESEKRA) for remote attestation which is focused on Edge computing in different subchapters. For example:

a.     7.1 Academic implications (if there are any)

b.     7.2 Managerial implications (if there are any)

c.      7.3 Future Work (if there are any)

Good Job and good luck!

Round 2

Reviewer 2 Report

I have no more comment